# Long-term additions of ammonium nitrate to montane forest ecosystems may cause limited soil acidification, even in presence of soil carbonate

5  Thomas Baer[1], Gerhard Furrer[2], Stephan Zimmermann[1], Patrick Schleppi[1]

[1]Forest Soils and Biogeochemistry, Swiss Federal Research Institute for Forest, Snow and Landscape Research (WSL), CH-8903 Birmensdorf, Switzerland
[2]Institute of Biogeochemistry and Pollutant Dynamics, ETH Zurich, CH-8092 Zürich, Switzerland

*Correspondence to*: Patrick Schleppi (patrick.schleppi@wsl.ch)

**Abstract.** Nitrogen (N) deposition decreased in the last decades in Europe, but in many cases remains higher than the critical loads, i.e. higher than what could be considered safe for biodiversity and ecosystem functioning . The main concerns about N deposition are eutrophication and acidification. In a long-term experiment (1994 to present) in a montane (1200 m a.s.l.) coniferous forest in Alptal, central Switzerland, we simulated increased N deposition by adding $NH_4NO_3$ to rainwater. This

treatment consisted of an additional N input of 22 kg ha$^{-1}$ y$^{-1}$ to the 12 kg ha$^{-1}$ y$^{-1}$ ambient bulk deposition rate or 17 kg N ha$^{-1}$ y$^{-1}$ throughfall rate. The treatment was applied simultaneously to a small catchment area and to plots in a replicated block experiment (n=5). The site has a carbonate-rich parent material and is thus not particularly at risk of acidification. Nevertheless, we examined soil acidification as affected both by ambient and experimentally increased N deposition. In the two and a half decades since the beginning of the study, nitrate ($NO_3^-$) and especially sulfate ($SO_4^{2-}$) concentrations

decreased in precipitation, while pH increased by slightly more than 1 unit. In the same time period a reduction in pH of the soil was measured. The exchangeable acidity in the soil increased, especially in the N-addition treatment. This was mainly observed on small mounds because the drier mounds are less well buffered than wet depressions. This trend, however, was limited in time as exchangeable acidity later declined again, to reach values not much higher than 26 years before. This was also the case in the N-addition treatment and can be considered a progressive recovery mainly due to the reduced acid inputs

and, at this site with a carbonate-rich sub-soil, to the biological cycling of base cations. The pH of the runoff from the experimental catchments decreased by 0.3 units, both in the control and under N-addition. Decreasing $Ca^{2+}$ and increasing $Al^{3+}$ and $Fe^{2+}$ concentrations in runoff are also showing that the recovery observed in the exchangeable soil acidity is not yet able to stop the slow acidification of water leaving the catchments. However, with the runoff water pH remaining above 7, this trend is not alarming for water quality or for the health of water bodies. Future monitoring will be necessary to see if and

when a recovery takes place in the soil and runoff pH.

# 1 Introduction

Due to anthropogenic activity, the transformation rate of inert $N_2$ into reactive N has doubled compared to pre-industrial times (Vitousek et al., 1997). The reasons for this influx of reactive N are increases in combustion of fossil fuels, widespread application of N fertilizer and increased cultivation of N-fixing crops (Galloway et al., 2003). Globally, close to 80% of the emissions of $NO_X$ and around 70% of the emissions of $NH_4^+$ result from human activity (Schlesinger & Hartley, 1992; Delmas et al., 1997). A large share of anthropogenic N emissions ends up in the atmosphere and can be transported over long distances and deposited in dry or wet form. Together with sulfate ($SO_4^{2-}$), $NO_X$ compounds are responsible for the acidity in atmospheric precipitation.

As large regions of the terrestrial biosphere are N-limited, increased N deposition can have significant effects on biological activity both below and above ground. In particular, forests in temperate regions are thought to be naturally N-limited (Vitousek & Howarth, 1991). Moderate increases of N deposition in forest soils have been shown to be correlated with increased mineralization rates, nitrification, and plant growth rates. Nitrification releases protons ($H^+$) into the soil solution (Equation 1).

$$NH_4^+ + 2\,O_2 \rightarrow NO_3^- + 2\,H^+ + H_2O \tag{1}$$

$NH_4^+$ taken up by plants corresponds to an exchange of this cation with $H^+$, which is then released into the soil (Galloway et al., 2003; Högberg et al., 2006). These biologically induced releases of $H^+$, together with the increased deposition of $H^+$ caused in part by $NO_X$ from the atmospheric precipitation, mean that soil acidification is a possible consequence of increased N deposition. This has been observed in a wide range of studies (Boxman et al., 2008; Bowman et al., 2008; Lu et al., 2009; Högberg et al., 2006; Lieb et al., 2011).

Denitrification, i.e. transformation of N from $NO_3^-$ to atmospheric $N_2$, on the other hand, is a sink for $H^+$ (Equation 2).

$$2\,NO_3^- + 10\,e^- + 12\,H^+ \rightarrow N_2 + 6\,H_2O \tag{2}$$

Besides the balance between inputs and outputs, the rate at which the soil acidifies depends on its buffering capacity. The dominant processes by which $H^+$ is buffered is either through biologically induced redox reactions, which can take place in decomposition of biomass, or through mineral weathering and cation exchange mechanisms on the surfaces of soil solids. These buffering mechanisms remove $H^+$ from the soil solution, reducing its acidity. Soil pH and the mineralogy of the soil determine which type of buffering process dominates. The dissolution of calcium carbonates is the dominant buffering process in calcareous soils (Equation 3).

$$CaCO_3 + 2\,H^+ \rightarrow Ca^{2+} + CO_2 + H_2O \tag{3}$$

At very low pH, hydrolysis and dissolution of Al and Fe complexes are the dominant buffering systems (Chadwick & Chorover, 2001).

If the rate of inflow of $H^+$ is higher than the rate of the buffering mechanisms, the soil will acidify. Soil acidification can be recognized by a number of factors including a reduction in soil pH over time indicating a higher concentration of $H^+$ in the soil. A further indicator is an increase in the proportion of acidic cations compared with the proportion of base cations in the

cation exchange capacity (CEC). As a consequence of decreasing pH, cations such as $H^+$, aluminum ($Al^{3+}$), iron ($Fe^{2+}$) and manganese ($Mn^{2+}$) replace base cations such as calcium ($Ca^{2+}$), magnesium ($Mg^{2+}$), potassium ($K^+$) and sodium ($Na^+$) from the cation exchange sites on the surfaces of mineral particles and organic matter in the soil (Amelung et al., 2018). This leads to an overall reduction in the base saturation (BS) and a reduction of the buffering capacity. Cations, which are being replaced from exchange sites or stem from dissolution reactions of minerals, enter the soil solution and can leach from the

soil. Runoff water from acidified soils shows higher concentrations of dissolved base cations and dissolved metals, especially $Al^{3+}$ and $Fe^{2+}$ (Warfvinge & Sverdrup, 1984). Soil acidifying processes can therefore be observed directly in the soil or indirectly in the chemical composition of the runoff water.

Soil acidification due to increased N deposition can have adverse impacts on the biosphere and on the surrounding environment. Possible effects on plants can be Al toxicity as a result of increased Al loads in the soil solution and increased

loads of Al adsorbed at the cation exchange sites. The depletion in base cations in the soil can also lead to nutrient imbalances affecting plant growth (Richter et al., 2007; Göransson & Eldhuset, 1995). Furthermore, the increased leaching of N, due to N saturation and increased nitrification rates, can affect water bodies downstream by inducing eutrophication. Increased N deposition is not the only possible driver of soil acidification. Precipitation with low pH values, commonly known as acid rain, is another source of $H^+$ to the soil. The low pH is mainly caused by high concentrations of sulfate ($SO_4^{2-}$)

and $NO_X$. Thanks to stringent emission regulations in Switzerland and in Europe in general (Convention on Long-Range Transboundary Air Pollution), $SO_4^{2-}$ concentrations in precipitation have declined and the pH of precipitation has increased over the last 30 years. However, achieving reduction in oxidized ($NO_X$) and especially reduced ($NH_3$) N emissions is more difficult, and N deposition continues to be an environmental problem. In Switzerland, a large share of the natural ecosystems receive N deposition above their critical load (Rihm & Achermann, 2016). In large parts of the world, the deposition of

reduced N also decreases less than that of oxidized N (Templer et al., 2022).

A considerable amount of research has been conducted on the effects of increased N deposition on weakly buffered soils such as alpine soils (Lieb et al., 2011), tropical soils (Lu et al., 2014), subtropical soils (Lu et al., 2009) and temperate non-calcareous soils (Boxman et al., 2008). Soil acidification has been observed in all those cases. However, the effect of increased N deposition on the biogeochemistry of a well-buffered system has not been examined to the same extent. In the

present study, the biogeochemical development of such a well-buffered forest soil in the Prealps of Switzerland was analyzed. A long-term N-addition experiment has been conducted at that site, during which a catchment area in a coniferous forest was subjected to $NH_4NO_3$ additions at a rate of 22 kg ha$^{-1}$ y$^{-1}$ compared to a control catchment receiving only ambient N deposition or throughfall at a rate of 12 kg ha$^{-1}$ y$^{-1}$ or 17 kg N ha$^{-1}$ y$^{-1}$ respectively. Atmospheric precipitation and runoff water from these catchments were regularly analyzed over the entire period of the experiment (Schleppi et al., 1998).

In this study we investigate the acidifying effects on the soil as a result of ambient N deposition in combination with N from the experimental addition of $NH_4NO_3$. Precipitation chemistry over time was used to test a first hypothesis:

*1) Atmospheric precipitation became less acidic over time due to lower levels of pollution. This should be observable as an increase in precipitation pH and a lower rate of soil acidification in the control.*

To test the effect of experimental N addition on the soil, samples from plots subjected to the increased N inputs as well as the controls were analyzed to examine a second hypothesis:

*2) Relative to the control, the N-treated soil (organic horizon and topsoil) experienced a stronger acidification, measured as a lower pH and an increase in total acidity.*

The site shows water movements through the soil mainly as preferential flow. This limits the contact between water and the bulk of the well-buffered Bw horizon. Soil acidification may thus affect the chemical composition of runoff water. Chemical data collected over more than 20 years from the experimental catchments were used to test a last hypothesis:

*3) Soil acidification can affect the pH and the chemical composition of runoff water even at a site with a well-buffered subsoil.*

## 2 Methods

### 2.1 Site description

The present study was conducted in the Alptal valley, in central Switzerland (47°03' N, 8°43' E), at 1200 m a.s.l. Flysch is the parent rock material, which is composed of sedimentary conglomerates with clay-rich schists. The main soil types are very heavy Gleysols. They contain on average 48% clay and have low permeability, leading to a water table close to the surface throughout the year (23 cm deep on average, Krause et al., 2013). The slope is about 20% with a west aspect. Soil characteristics differ depending on the prevailing topography. At a scale of typically 10 m, the topography shows a mosaic of mounds and depressions. The mounds are characterized by a deeper water table and have a profile with mor (raw humus), Ah and an oxidized or partly oxidized Bw horizon. The pH ranges from 3 to 4 at depths down to ~10 cm. In the depressions, the water table can reach the surface, leading to waterlogged conditions. The profile consists of an anmoor (muck humus) topsoil, and an almost permanently reduced Bg horizon (Hagedorn et al., 2001b). The pH lies between 4.5 and 6 at depths of 0 to ~10 cm. Depending on the locations, annual net nitrification rates (down to a depth of 15 cm) are partly positive and partly negative (N immobilization). Overall, the net nitrification rate is not significantly different from zero (Hagedorn et al., 2001b).

The climate of the study is cool and wet, with an average annual temperature of 6°C and average annual precipitation of 2300 mm, reaching a maximum of 270 mm in June and a minimum of 135 mm in October (averages from 1994 to 2015). A naturally regenerating mature Norway spruce stand (*Picea abies* (L.) Karst.) is present on the site, mixed with 15% silver fir (*Abies alba* Mill.). The trees are up to 270 years old, have a dominant height of ca. 30 m and a low leaf area index (LAI) of 3.8 (Schleppi et al. 1999b). As waterlogged soil conditions inhibit growth in depressions, the trees grow on the mounds,

accompanied by a ground vegetation dominated by *Vaccinium myrtillus* and *Vaccinium vitis-idaea*. In the depressions, ground vegetation is dominated by *Caltha palustris*, *Petasites albus*, *Poa trivialis* and *Carex ferruginea*. Rooting depth of all plants is limited to 10–25 cm depending on the micro-topography (Schleppi et al., 1999b). At the beginning of the

experiment atmospheric deposition of inorganic N in bulk and throughfall was 12 and 17 kg N ha$^{-1}$ y$^{-1}$, respectively (Schleppi et al., 1998).

## 2.2 Experimental catchments and N addition

An N addition experiment was conducted at the Alptal site to assess the effects of atmospheric N deposition. The design of the experiment (Fig. 1) was two-fold. First, $NH_4NO_3$ dissolved in rain water was added by sprinklers to a small headwater

catchment and compared to a control catchment in a paired-catchment design (Schleppi et al., 1998). The added rainwater to both catchments only leads to a mild increase of 7% on average to the annual ambient precipitation.

Second, small plots received the same treatment as the catchments at the same time but in a replicated block design (n=5) (Hagedorn et al., 2001a). Each plot had a sprinkler at its center, spraying either rain water (control) or rain water with added $NH_4NO_3$ (N-addition treatment). Blocks (pairs of plots) were located so as to cover the variability of the topographical

features: two blocks were set on mounds and three in depressions. The small plots were mostly just outside the catchments, but in two cases N-addition plots were inside the corresponding catchment (Fig. 1). Combining both designs enabled us to study soil processes in more detail in the replicated plots while we could also obtain an integration of relevant processes at the larger scale of the catchments, encompassing the different topographic features as well as a larger number of trees. The two forested catchments, each approximately 1500 m$^2$ in area, were delimited by trenches leading to the gauging

stations.. These trenches were dug one year before measurements began and reach into the gleyic sub-soil while capturing the sub-surface water flow (Schleppi et al., 1998). No change of the vegetation along the trenches has been observed that would indicate enhanced drainage from the soil along the trenches. Catchment 1 (control) has a convex shape in the upper region and did thus not require trenches there (Fig. 1). Both catchments feature a similar topography with similar proportions of mounds (approx. 40%) and depressions (approx. 60%). Water level is measured continuously in V-notch weirs (custom

made) at the bottom of the catchments. Water discharge is calculated from these measurements and is averaged over 10 min periods (Schleppi et al., 1998). Both catchments always showed a rapid response to rain events and exhibited similar runoff peaks (Schleppi et al., 2004).

One year of measurements with weekly water analyses served as a calibration period before N-addition treatment began. Increased N deposition was then simulated in one of the catchments by applying $NH_4NO_3$ dissolved in rainwater with

permanently deployed sprinklers. Rainwater was collected on a polyethylene sheet (300 m$^2$) spread outside the forest and channeled into a water tank (2000 L). N addition occurred automatically when the tank was full, in small but frequent amounts: approximately 200 times per year for an average of 22 kg N ha$^{-1}$ y$^{-1}$ additional N input to the ambient deposition rate of 12 kg N ha$^{-1}$ y$^{-1}$. In a paired-catchment design, the other catchment served as a control and received the same amount

of rain water without any added N resulting in a N input of 12 kg N ha$^{-1}$ y$^{-1}$. During winter, N treatment was done with a backpack sprayer as occasional applications of a concentrated NH$_4$NO$_3$ solution on the snow cover.

In June 2009, 15 trees per forested catchment, all with a diameter at breast height of >20 cm, were girdled by removing bark and cambium around the stem (Krause et al., 2013). One year later, the girdled trees became infested by bark beetles (*Ips typographus*) and were felled in August 2010 to prevent the spread. This represented a heavy selective cutting (38% of the total basal area of all trees >10 cm diameter at breast height) affecting both catchments equally. The branches (with needles still attached) were left on site, while the boles were removed by helicopter. This disturbance did not affect soil temperatures (7.1°C on average) nor soil water depths (23 cm on average) (Krause et al., 2013).

## 2.3 Sampling and analyses

Precipitation was measured and sampled weekly from an open area approximately 150 m west of the forested catchments. Two 0.05 m$^2$ rain gauges (custom made) were used, except in the winter when only one was heated in order to let the snow melt. These gauges consist of a polypropylene funnel mounted on top of an opaque PVC tube with a polyethylene 5-L collection bottle inside. Water samples were proportionally combined fortnightly and filtered (0.45 µm). NH$_4^+$ was analyzed by flow injection, anions by ion chromatography and cations by induced-coupled plasma atomic emission spectrometry (ICP-AES) (Schleppi et al., 1998). The pH was measured with a Metrohm 654 pH-meter (Metrohm, Herisau, Switzerland). Its electrode was filled with a low-concentration electrolyte (0.55 M KCl) to adapt to the low ionic strength of precipitation samples. Further, 10% of the same 0.55 M KCl were added to each sample prior to measurement. Stable readings were difficult to achieve with precipitation water. For this reason, samples were bubbled with N$_2$ to favor degassing and readings taken after a fixed time of 6 minutes. To check the long-term consistency of the pH measurements, precipitation samples were proportionally pooled over periods of 3 months and frozen during the entire experiment. In 2016 these samples were melted, measured again with the current 654 pH-meter (Metrohm, Herisau, Switzerland) and compared to values calculated from the H$^+$ concentrations derived (by weighted averages) from the original measurements done just after collection during the experiment.

Runoff water was collected proportionally to the discharge (an aliquot per 200 or 300 L discharge) and bulked over 2 weeks (Schleppi et al., 1998; Schleppi et al. 2006). These samples were analyzed with the same methods as for precipitation water. For their pH measurement, a Metrohm 691 pH-meter (Metrohm, Herisau, Switzerland) was used, with a (normal) KCl 3 M electrolyte.

Soil samples were taken from the 10 plots (Fig. 1) at irregular intervals of several years (1996, 2007, 2014, 2015, 2016 and 2022). Samples were taken as 3 cores per plot (5 cm diameter). They were immediately refrigerated, brought to the lab, separated by soil layer and bulked over the 3 cores per plot. Stones and roots were removed and the samples were then dried at 65°C. Samples taken in 1996 had been frozen and were dried and measured only in 2017. Samples taken in 2007 had been dried and were measured also in 2017. In other years, measurement followed drying within one month at most. The O and A horizon were included in our analyses. The Bw horizon receives almost no N from the experimental addition, as shown by

[15]N labelling (Providoli et al., 2006). As the Bw horizon is well buffered by the presence of carbonate, it was not included in our analyses on acidification.

10 g of each soil sample were suspended in 20 ml 0.01 M $CaCl_2$ for half an hour. For soil samples high in organic matter content, only 5 or 2.5 g were used. The pH was subsequently measured with a Hamilton 9.99 electrode on a Metrohm 691 pH-meter (Metrohm, Herisau, Switzerland). The effect of the soil amount was assessed by weighing 2, 4, 6 and 8 g from each horizon and suspending all in 20 ml 0.01 M $CaCl_2$. The $H^+$ concentration varied by less than 3 % between the different soil amounts, which means that the difference in pH due to varying weight of soil was negligible. For this reason, no pH correction for soil weight was needed.

Total soil acidity (defined as the sum of exchangeable $Al^{3+}$ and $H^+$) was extracted by suspending 5 g of each soil sample (in duplicate) in 50 ml 1 M KCl and mixing them in an overhead stirrer for one hour. If the soil had a high organic matter content, or if only a small amount of soil sample was available, only 2.5g were used instead of 5 g. The extract was then filtered through 5893 pleated filters. Total acidity and exchangeable $Al^{3+}$ were measured using an automated sampler (814 USB Sample Processor, Metrohm). The exchangeable $H^+$ was calculated by subtracting the amount of exchangeable $Al^{3+}$ from the total acidity. Reference samples were measured at regular intervals, in order to check for drift. The samples were measured in sequence from highest to lowest pH, in order to minimize possible carry over between the single measurements. Samples with a pH of 3.55 and below were diluted with the equal volume of 1 M KCl. Samples with a pH higher than 6.0 were not measured for soil acidity. The resulting acidity values were then corrected for weight and dilution wherever necessary.

Exchangeable cations were extracted with 1 M $NH_4Cl$ (soil:solution ratio: 1:10) during 1 hour and measured by ICP-AES. The effective cation exchange capacity ($CEC_{eff}$) was calculated by summing up the charge equivalents of exchangeable H, K, Ca, Mg, Al, and Fe. The base saturation (BS) equaled the relative fraction of K, Ca, and Mg of the $CEC_{eff}$.

## 2.4 Statistics

Changes in precipitation chemistry were estimated using linear models with the lm function in R, with time and seasonality as explanatory variables. Seasonality was tested using a sinusoidal approach, testing for both the sine and the cosine of the yearly cycle.

Linear mixed-effects models were used to estimate the effects of the N treatment, time, and their interaction on soil acidity and pH. Models were fitted with the lme function from the nlme R package version 3.1-162 (Pinheiro et al., 2023). Treatment (control or N addition) and time were used as fixed effects, as well as a second-degree polynomial for time to allow for slopes changing during the course of the experiment. Replications (pairs of plots) were considered a random effect allowing for random intercepts and slopes by treatment, as the same locations were sampled over the years and cannot be assumed to be independent. The interaction between treatment and time was also tested, but was subsequently removed as no significance

could be shown. The dependent variable soil acidity was log-transformed to correct for the unequal variance between replications due to the topography.

To test for differences in total acidity between topography, three fixed effects were used: treatment, topography (mound or depression) and interaction between the treatment and topography. Allowing for random slopes for treatment did not let the model converge, so only random intercepts were used. Separate linear mixed-effects models (with identical structures) were fitted for the A and the O soil horizon.

Runoff water chemistry changes were estimated using the same approach as with precipitation with the following explanatory variables: time, seasonality, treatment (control or N addition) and interaction between treatment and time. The interaction was tested by introducing a pseudo variable for both treatments, accounting for reference period during the first year before treatment start and then increasing for N treatment and decreasing for control. In order to represent only the interaction, the sum of the pseudo variable had to be zero across treatments and also across time points.

The time series of pH in precipitation and in runoff were available in fortnightly steps. In a first approach, we tested these series for auto-regression with the acf function from the stats package (R Core Team, 2013) of R (version 4.1.2).

We further tested the short-term correlation between pH in precipitation and runoff. For this purpose, we filtered for data between 21 May and 28 October, to avoid data obtained in the presence of snow cover. For each year, we calculated the deviations compared to the average of that year, thus removing the long-term trends and keeping only the short-term variations. Based on this data, correlations were calculated with a shift between zero and nine time steps from the first to the second series.

## 3. Results

### 3.1 Precipitation chemistry

Overall, precipitation rates in the Alptal showed no significant change over time between the beginning of the experiment in April 1994 and the latest available measurements (2019). Annual precipitation averaged 2130 mm. The bulk precipitation data showed a significant increase in pH of the precipitation ($P < 0.001$; Table S1, Fig. 2). At the start of the experiment, average pH was above 5 and increased to 6.8 by the end of 2019. The increase in pH took place at an average rate of 0.068 per year (SE $\pm$ 0.003). Occasional very high values (up to pH 8) were observed, which are linked to atmospheric conditions that result in dust originating from the Sahara being transported over the Alps.

In the same time period, $SO_4^{2-}$ and $NO_3^-$ concentration in the precipitation both showed a significant decrease ($P < 0.001$; Table S1, Fig. 2). Since 1994, the average $SO_4^{2-}$ concentration more than halved at a rate of 0.029 mg $l^{-1}$ per year (SE $\pm$ 0.003). The $NO_3^-$ concentration also decreased over time, but not as strongly as the S deposition (Table S1, Fig. 2). The concentration of $NH_4^+$ did not change significantly during the years of monitoring and remained at an average level of ca. 0.3 mg $l^{-1}$. All measured variables showed a significant seasonality ($P < 0.001$; Table S1, Fig. 2), with pH showing a minimum around July and $SO_4^{2-}$, $NO_3^-$ and $NH_4^+$ showing maximum values in June, May and April respectively.

## 3.2 Soil

Soil pH decreased significantly over time for both the N-treated soil and the control soil ($p < 0.001$; Fig. 3, Table S2). Neither the rates of decrease in pH nor the average values differed significantly between the two treatments. The O horizon showed slightly lower pH values with a stronger decrease in pH over time than the A horizon.

Total acidity showed a significant ($p < 0.001$; Fig. 3, Table S2) non-linear change over time for both the control and N-treated soil. It initially increased over time, peaking around 2012 and then decreasing continuously. The same characteristics for total acidity were observed for both soil horizons, however with lower values for the A compared to the O horizon.

The variation in total acidity between replicates (measured values) can partly be explained by topography. There were 5 replicates for each of the sampling campaigns in the years 1996, 2007, 2014, 2016 and 2022 as described in the methods. Two of the replicates were located on mounds and three in depressions. Total acidity on the mounds differed significantly from total acidity in the depressions ($p = 0.006 / 0.012$ for O and A horizon respectively; Fig. 4, Table S2). The interaction between topography and treatment was marginally significant for the O horizon and significant for the A horizon ($p = 0.070 / 0.023$ for O and A horizon respectively), so that N-treated mounds were the most acidic.

A difference in proportions of exchangeable cations between the control and N-treated soil was observed when examining the soil from mounds and depressions separately for both the O and the A horizon (Fig. 5). The N-treated soil showed a higher percentage in acidic exchangeable cations (shown in red in Fig. 5), mainly $Fe^{2+}$, $Al^{3+}$ and $H^+$, than the control. The control accordingly exhibited a higher percentage of exchangeable base cations (shown in blue in Fig. 5), mainly $Ca^{2+}$ and $Mg^{2+}$. The acidic exchangeable cations in the N-treated soil make up close to 40 % and 50 % in the O and A horizons respectively, whereas in the control soil they only make up ~ 15 % and ~ 10 %. The difference in exchangeable cations between depressions and mounds were observed, with the depressions showing less than 10% of the exchangeable acidic cations. $Ca^{2+}/Al^{3+}$ ratios ranged from over a 1000 in the depressions to about 1.4 on the mounds.

## 3.3 Runoff water

The concentrations of all anions and cations (except for $Fe^{2+}$) in the runoff water changed significantly over time ($p = <0.001$) for the control catchment. $Al^{3+}$ and pH showed marginally significant interactions ($p = 0.073$ and $p = 0.092$) and $Fe^{2+}$, $SO_4^{2-}$ and $Ca^{2+}$ showed highly significant interactions between treatment and time (Table S3 and Fig. 5). The pH decreased for both catchments with two distinct periods of relative higher pH from 2010 to 2012 and 2017 to 2020. $Al^{3+}$ and $NO_3^-$ increased in concentration in the runoff water from both catchments, while showing a significantly higher rate of increase for the N-treated catchment (Fig. 6 and Table S4). $Fe^{2+}$ concentrations increased significantly in the runoff from the N-treated catchment, but stayed stable in the control. A significant decrease in concentration of $SO_4^{2-}$ and $Ca^{2+}$ was observed for both catchments. The decrease in concentration in the runoff water from the N-treated catchment was significantly lower compared to that of the control catchment. All anion and cation concentration showed significant seasonal trends (Fig. 6 and Table S4).

The observed $NO_3^-$ concentration showed a distinct non-linearity in the runoff water from the N-treated catchment. For this reason, a $5^{th}$-order polynomial was included in the model. The water from the N-treated catchment showed a sudden increase in concentration of $NO_3^-$ starting around 1995 and increasing constantly till reaching a maximum in 2010, after which it started to decreased. At the end of 2018 the concentration reached a minimum, then started to increase slowly again. In comparison, the control did not show much variation over time, other than a slight decrease after 2010. Two dates are highlighted in the $NO_3^-$ plot in Figure 5: March 1995, indicating the beginning of the $NH_4NO_3$ additions to the N-treated catchment, and June 2009, which indicates the girdling of select trees prior to their felling in August 2010.

The auto regression between precipitation and runoff showed mainly that correlations decreased from 0.6 (precipitation) or 0.5 (runoff) at a lag of 1 fortnightly period to reach a first minimum at a lag of 12-14 fortnightly periods, then increasing again up to 22-27 fortnightly periods, corresponding to a yearly cycle.

Runoff water pH and precipitation pH showed no significant short term correlation to each other for the same time period for either the control or the N-treated catchment, with correlation coefficients of -0.081 and -0.078 respectively. The correlation was also not significant when the time periods were offset by 1 to 9 timesteps (1 timestep equals 2 weeks), with coefficients ranging from -0.184 to 0.090 for the control and -0.108 to 0.122 for the N-treated catchment.

## 4 Discussion

### 4.1 Atmospheric deposition

Based on general trends in the last decades, our first hypothesis was that our experimental site is subjected to decreasing atmospheric acid inputs. From 1994 to 2019, precipitation sampled at the Alptal site has indeed become significantly less acidic. This was clearly visible in the increase in pH and in the decreasing concentrations of $SO_4^{2-}$ and, to a lesser extent, $NO_3^-$ of precipitation water. The $NH_4^+$ concentrations showed no significant change over time. These findings are consistent with other sites in Switzerland which have reported, since the mid-1990's, partly significant decreases in $NH_4^+$ and mostly significant decreases in $NO_3^-$ deposition (Thimonier et al., 2019). Data from our research site are also in line with average decreases measured since 1988 by the national air pollution monitoring in wet deposition of $SO_4^{2-}$ (-79%), $NO_3^-$ (-36%) and $NH_4^+$ (-20%) (BAFU, 2022). These trends are clearly the result of measures taken to reduce anthropogenic emissions of $SO_4^{2-}$, $NO_X$ and $NH_3$, respectively. Similar trends are generally observed in Europe (EMEP, 2021). In a world-wide study of time series from forest and grassland watersheds, Templer et al. (2022) also found a general reduction in bulk $NH_4^+$, and especially $NO_3^-$ deposition. This means that reduced N compounds become progressively more important relative to oxidized N in atmospheric deposition.

The observed increase in pH in our precipitation samples was strong. As a linear trend of measurements from 1994 until 2019, pH increased by 1.63. Re-measuring frozen samples gave higher pH values for old samples, resulting in a weaker increase of 1.18 in the same 25 years. The observed trends over time in pH from original samples and from the

measurements of the frozen samples were not identical but also not significantly different (p = 0.075). It is not possible to determine which of these values is more accurate. Despite calibrating the pH-meter before every fortnightly measurement, we cannot completely exclude that there could be a long-time trend due to the aging of the pH-meter and electrodes. We cannot either exclude that a slow degassing took place in the frozen samples, leading to a loss of dissolved $CO_2$ and thus to a slight pH increase, even if the size of the bottles (mostly 250 - 1000 mL) does not favor degassing. Further, some buffering effects may lead to losses of $H^+$ during the mixing of fortnightly samples into quarterly composites, resulting in some differences compared to the pH calculated from the fortnightly measurements (by the weighted average of $H^+$ concentrations). Even with these uncertainties, we can conservatively conclude that $H^+$ inputs by precipitation decreased by one order of magnitude within the last 25 years. This is also in line with the trends observed over Switzerland, with a gain of more than 1 pH unit over the last 35 years (BAFU, 2022). As stated in this national report, the cause of this pH increase has to be sought in the above-mentioned decrease in emissions of N and especially S oxides.

## 4.2 Soil acidification

The pH of precipitation, i.e. its $H^+$ concentration, is not the only factor leading to soil acidification.. Soil acidification as a consequence of increased N deposition has been established in a wide range of studies (e.g. Lu et al., 2014; Lieb et al., 2011; Bowman et al., 2008). This leads to our second hypothesis that our N-addition treatment contributes to soil acidification. $NO_3^-$ can have an acidifying effect if leached together with base cations, leading to their replacement by $H^+$ or $Al^{3+}$ on the exchange complexes in the soil. $NH_4^+$ also has an acidifying effect if it is nitrified (Eq. 1) or taken up by plants in exchange with $H^+$. These two N ions represent the main potential of soil acidification from ambient precipitation. Adding them as N treatment simulated a strong increase of this potential in our experiment. As explained in a previous article (Schleppi et al., 2017), $NO_3^-$ leaching quickly increased as a result of the N-addition. This was interpreted as a short-term effect linked to fast, preferential water flow through the top-soil, leading to the loss of approximately 10% of the added N. After several years of treatment, we observed a further reduction of $NO_3^-$ retention, down to approximately 70%, which we consider a sign of progressive N saturation of the ecosystem. Finally, a peak of $NO_3^-$ leaching was observed in the 5-6 years after girdling and subsequent felling of part of the trees. This peak was clearly more pronounced in the N-treated catchment than in the control and this was interpreted as a result of the decreased N demand of the trees. Since $NO_3^-$ leaching increased in the N-addition treatment (Fig. 6; see also Schleppi et al., 2017), we can ascertain that acidification occurred in part due to the acidifying potential of $NO_3^-$. This can explain the stronger increase in soil acidity observed on the mounds under N-addition compared to the control (Fig. 4, 5) and can also partly explain that $Ca^{2+}$ concentrations decreased less (as a result of buffering processes) in N-treated catchment than in the control (Fig. 6). The contribution of $NH_4^+$ is more questionable because the slightly increased N uptake by the trees (Krause et al., 2012) can also be in the form of $NO_3^-$. Further, net nitrification rate is minimal or even negative at times in the soil at our site (Hagedorn et al., 2001b). The added $NH_4^+$ is well retained in the ecosystem and abiotic fixation is believed to be a major process of retention (Providoli et al., 2006), which can limit its acidifying effect.

It is quite surprising that the total acidity appeared to decrease in the N-addition treatment at the same time as in the control, in spite of the much higher load of $NO_3^-$ and $NH_4^+$. We suggest that most of the acidifying potential of these ions is not expressed on this site. $NO_3^-$ leaching occurred, but also in the control (Fig. 6) and mostly by preferential flow of $NO_3^-$ in precipitation rather than by leaching of $NO_3^-$ from nitrification in the soil (Hagedorn et al., 1999). And, as explained above, nitrification and plant uptake of $NH_4^+$ are limited by the (partly abiotic) immobilization of this ion in the soil. This can explain the observed recovery after 2010, with a decrease of the exchangeable acidity ($H^+ + Al^{3+}$). This tendency is not reflected in the pH, but this can be explained by the losses of acidic cations from the exchange complexes into the solution during the last decade. Even if soil pH does not yet show signs of recovery, our first hypothesis of a decrease in acid inputs and in soil acidification was confirmed, and even surpassed considering the observed recovery in exchangeable acidity. As shown by our measurements (Fig. 5), the cation exchange sites of the topsoil of the depressions is buffered by a high saturation in base cations. However, on the mounds the saturation is lower, which can be explained by an increased leaching of base cations as water moves deeper into the soil and laterally towards the depressions. As a result, the lower buffering capacity of the mounds was further reduced by N-addition treatment. This is in line with our second hypothesis of an acidifying effect of the added $NH_4NO_3$. With time, however, the exchangeable acidity decreased again.

Besides the pH increase in precipitation and the fact that $NO_3^-$ and $NH_4^+$ only partly express their acidifying potential, biological element recycling certainly also plays a role in the observed recovery. Most studies on soil acidification were conducted on soils that lack, or no longer contain carbonate (See historical review by Grennfelt et al. (2020) and, for a Swiss perspective, Blaser et al. (2008)). In contrast, the subsoil of our site contains carbonate. It is thus a large source of base cations for plant roots. With the uptake of base cations from the lower soil horizons and their return to the forest floor as litter, this biological recycling acts against acidification and can be considered as a further buffering mechanism. This appears to be effective enough even if the gley horizon of our site features reducing conditions (Hagedorn et al., 2001b), which makes the soil unfavorable for deep rooting, i.e. for the ability of plants to access deep soil layers rich with base cations. This biological recycling of base cations from tree roots to canopy and back to the forest soil as plant litter is also a mechanism by which base cations can be redistributed between topographical features, i.e. from the base-cation richer depressions onto the more base-cation depleted mounds.

A further mechanism acting against soil acidification is denitrification (see Equation 3). Due to the low redox potential of the soil at our site, denitrification could be measured (Mohn et al., 1999) and showed a rate in the order of magnitude of 1 kg N ha$^{-1}$ y$^{-1}$.

### 4.3 Runoff water chemistry

Besides the replicated plots, our N-addition experiment is also conducted in a paired-catchment design. As expected, the most prominent effect of the treatment was an increase of $NO_3^-$ leaching out the N-addition catchment. We found that leaching peaked during approximately 5 years after the girdling and subsequent cutting of some of the trees in 2009/2010. A smaller peak was also measured in the control, where the same girdling and cutting was performed. These results were

discussed in detail in Schleppi et al. (2017) and the peak following girdling and cutting was interpreted as a result of reduced root uptake. For the present study, our third hypothesis was that ions other than just $NO_3^-$ would exhibit changes over time and be affected by the N addition.

At the beginning of the experiment, water sampled at the weirs of the small catchments showed a pH around 7.5. With time, it decreased to approximately 7.2. Two relatively sudden increases occurred at the beginning of 2010 and 2017. They could not be traced to any change in the measurements such as a change of electrodes. For the first period (2010-2011), the higher pH may be an effect of the reduced uptake of base cations by roots after girdling some trees on both catchments, which means less release of $H^+$ by these roots after they had used up their starch reserves within a few months (Krause et al., 2013). In the years 2017 to 2019, the pH measured in precipitation (with a different pH-meter) were also higher than the long-term trend. The two periods of higher pH values also did not correlate with soil frost, which occasionally occurs at this site (Stähli, 2017). Furthermore, the absence of any short-term correlation between runoff pH and precipitation pH in the same or in the previous 2-week periods is in contrast with what was previously observed for $NO_3^-$ concentrations (Schleppi et al., 2017). While the $NO_3^-$ signal is partly transmitted through the soil by preferential water flow, the different behavior of the pH can be explained by buffering reactions in the soil. Cation exchange especially is much more efficient than anion-exchange, thus completely decoupling these two time-series.

The decrease in pH was also measured in a third catchment nearby, which showed a decrease in pH of approx. -0.018 per year. This third catchment was not included in this study as it consists of fallow land and thus is not comparable from a vegetation point of view to the N treated and control catchment areas in the forest.

The overall tendency of declining pH was accompanied by a decrease in $Ca^{2+}$ and an increase in $Al^{3+}$ and $Fe^{2+}$ concentrations. This shift from base cations to acid cations is a clear sign of a progressive acidification of the soil, more precisely of the hydrologically active layers of the soil, since the bulk of the Bw horizon is practically impermeable. Unlike the exchangeable acidity in the soil, no sign of recovery was observed in the chemical composition of runoff water. Since the exchange complexes of the soil lost acid cations during the last decade, it is understandable that recovery processes are not synchronous. Overall, the signals confirm our third hypothesis that soil acidification translates into the composition of runoff water even on a site with a well buffered subsoil. The sequence of recovery, however, is not the same as often observed in forests that were subjected to strong soil acidification after high S deposition. In such highly impacted systems, the storage and subsequent release of sulfur plays a dominant role and can delay the recovery of the soil (Ahrends et al., 2022) more than that of surface waters (Lawrence et al., 2015).

**5 Conclusions**

The Flysch parent material of the Alptal soil contains calcium (and magnesium) carbonate. When an N-addition experiment was started in 1995, we did not expect to see any acidification effects as a result of the added $NH_4NO_3$. After years of monitoring the pH both in precipitation and in runoff water from our small experimental catchments, we noticed opposite

trends: less acid precipitation but a declining pH in the runoff. For the present study, we used older and newer soil samples to examine the underlying processes. The results show that the observed trends can be interpreted as a temporal shift in a recovery process initiated by the (legally imposed) abatement of emissions of N and especially S oxides. The first effect is a reduction of the acidity in precipitation. At first the weakly buffered mounds of our site acidified. The N treatment (addition of roughly twice as much N as in the ambient deposition) had a significant effect on the acidification. Later, the reduction in

acid inputs from precipitation allowed base cations to replace part of the exchangeable acid cations in the soil, even under continuing N addition. This process, which is certainly linked to the biological cycling of base cations, takes time but may lead to a recovery of the soil pH in the future. A pH decline was also observed in the runoff water of our small experimental catchment. However, the pH remained above 7 and this trend is not alarming for water quality or for the health of water bodies. Future monitoring will be necessary to see if and when a recovery takes place in the soil and runoff pH.

**Author contribution**

As a principal investigator, PS was significantly involved in the set-up and running of the field experiment. TB provided most soil analyses as well as the re-measurement of pH from frozen samples for the present study. TB and PS wrote the manuscript with contributions of the other authors.

**Competing interests**

The authors declare no competing interests.

**Data availability**

Upon request, data obtained during the present study are available from the corresponding author.

**Acknowledgements**

The authors thank former graduate students who worked on this long-term project, including taking water samples and measuring pH. Among them, we especially thank Frank Hagedorn who provided the older soil samples. Most chemical analyses were done at the central laboratory of the Swiss Federal Institute for Forest, Snow and Landscape Research under the direction of Daniele Pezzotta. We also thank Sandra Angers-Blondin for support and advice with R programming and modeling and for proofreading the manuscript for final submission.

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

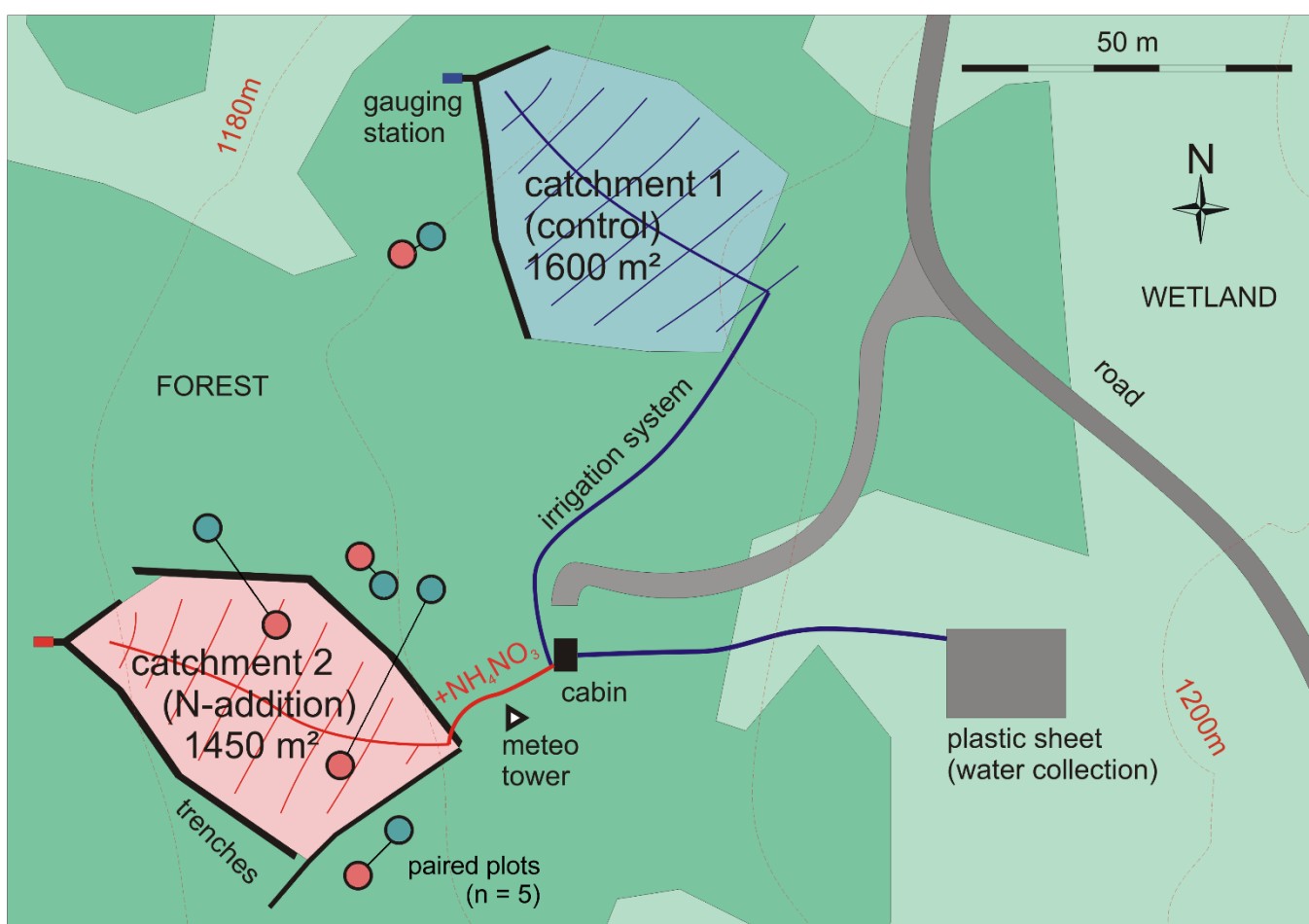

**Figure 1: Map of the experimental setup showing the small experimental catchments in the Alptal forest. Catchment 1 has a convex shape in the upper region and did thus not require trenches there. The same treatments as in the catchments (control or N addition) were also applied to small plots in a replicated design (n=5). Blue circles indicate control plots and red circles indicate N addition plots (replications, shown here as lines connecting plots). The location of each plot was chosen so as to minimize the**

545 **differences (topography, vegetation, light) within the pairs (replications) but to cover the variability of the site between them.**

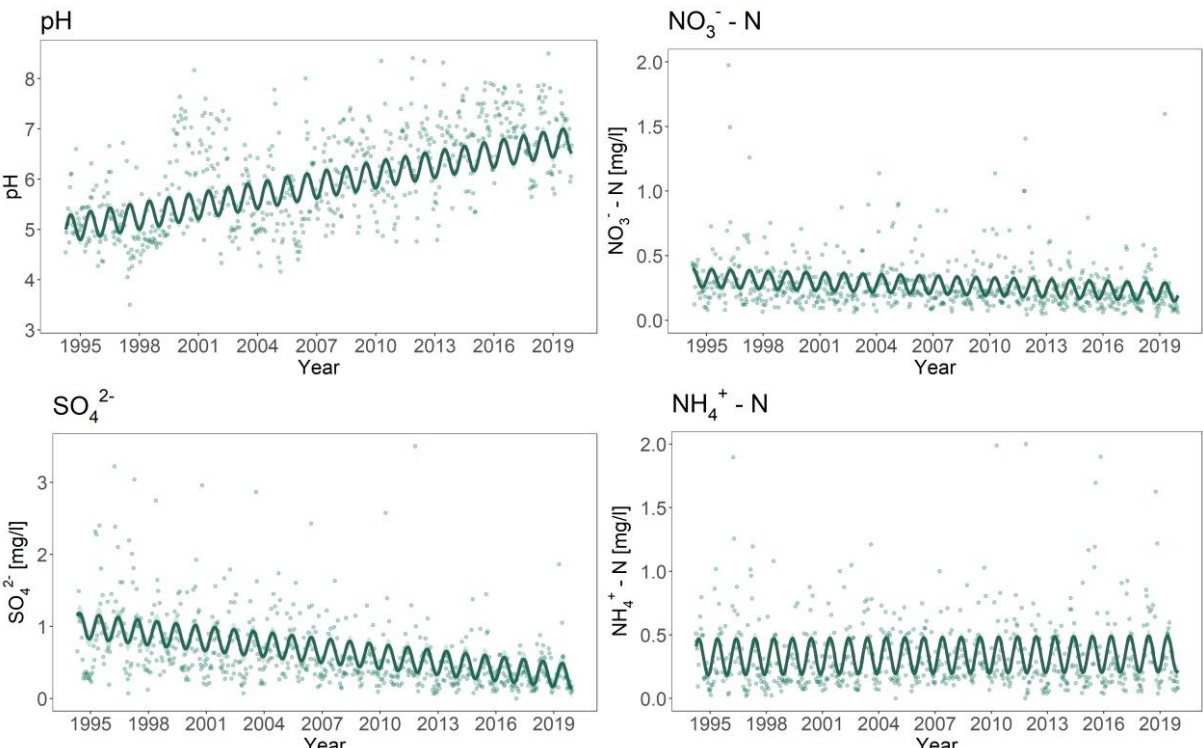

**Figure 2: Cation and anion concentration in precipitation from the study site in the Alptal as a function of time. The points are measured values with a timestep of 2 weeks. The continuous solid lines are the predicted values calculated with linear models. 8 and 2 outliers were excluded from the graphs for SO$_4$$^{2-}$ and NH$_4$$^+$ respectively.**

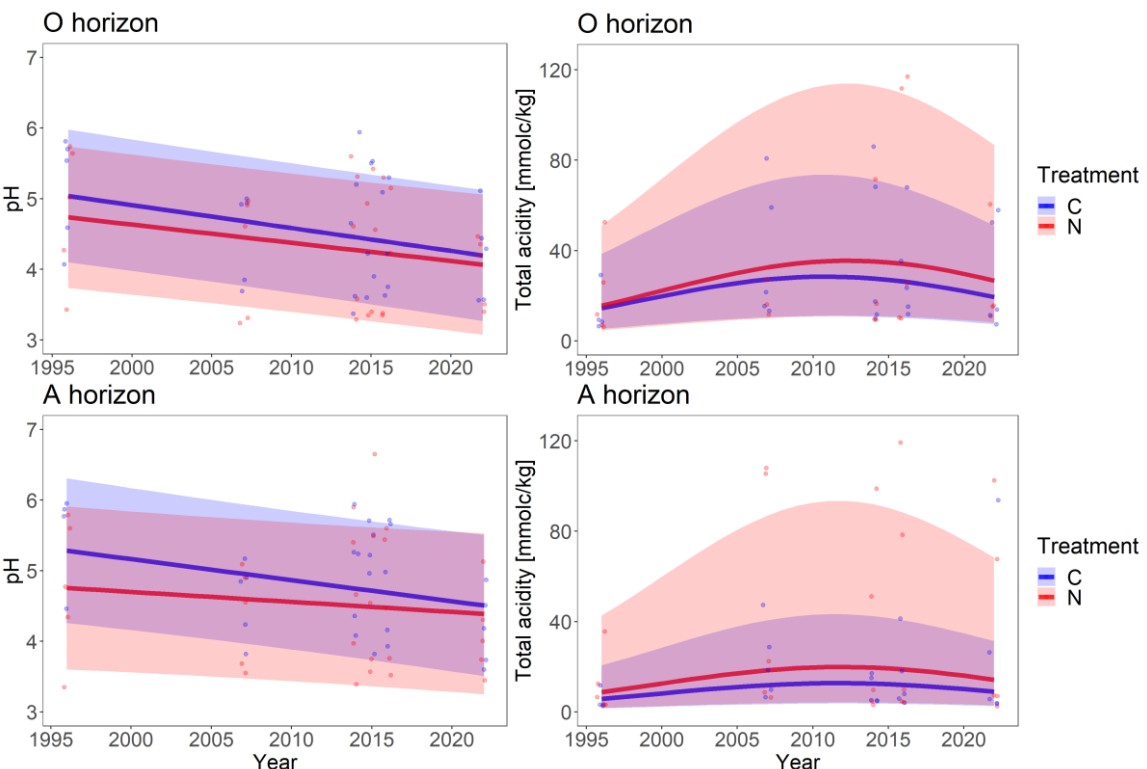

**Figure 3: Soil pH and total acidity predictions for O and A horizon based on linear mixed effects model (continuous line with confidence intervals) as a function of time, calculated with measured values from the years 1996, 2007, 2014, 2016 and 2022 (individual points). 4 highest points were excluded from pH plot for O horizon. Modeled total acidity values were back transformed from log scale. 2 outliers were excluded from O horizon plot for total acidity.**

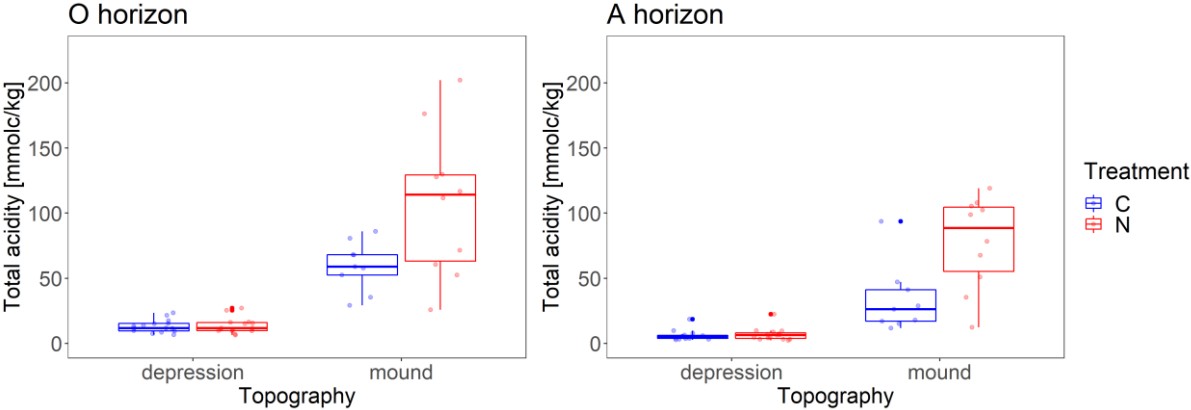

**Figure 4: Observed total soil acidity values from all years for O and A horizon for both topographical features (mounds and depressions).**

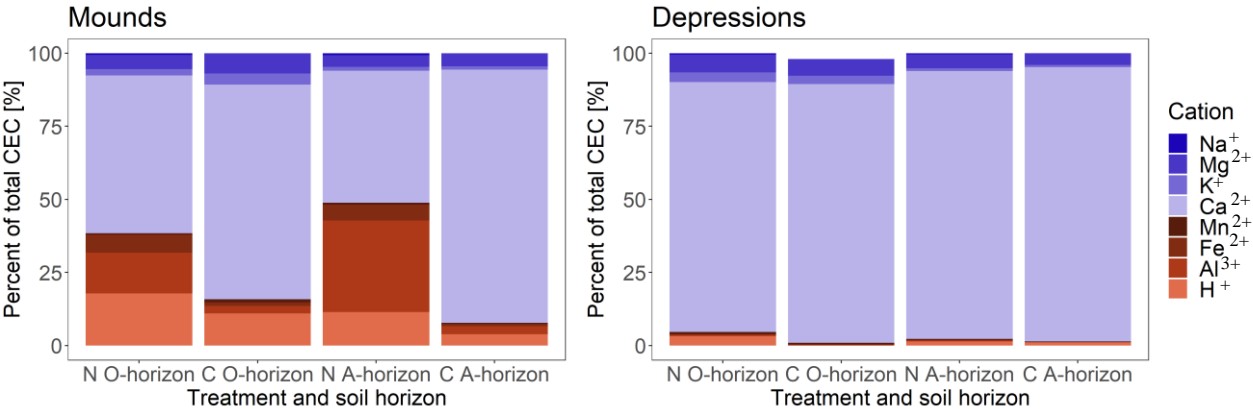

**Figure 5: Exchangeable cations in proportions of total cation exchange capacity (CEC) for the year 2016. (Pb an Zn were excluded as their concentrations was negligibly small or zero for most replications). Acidic cations are indicated by the color red and base cations are blue.**

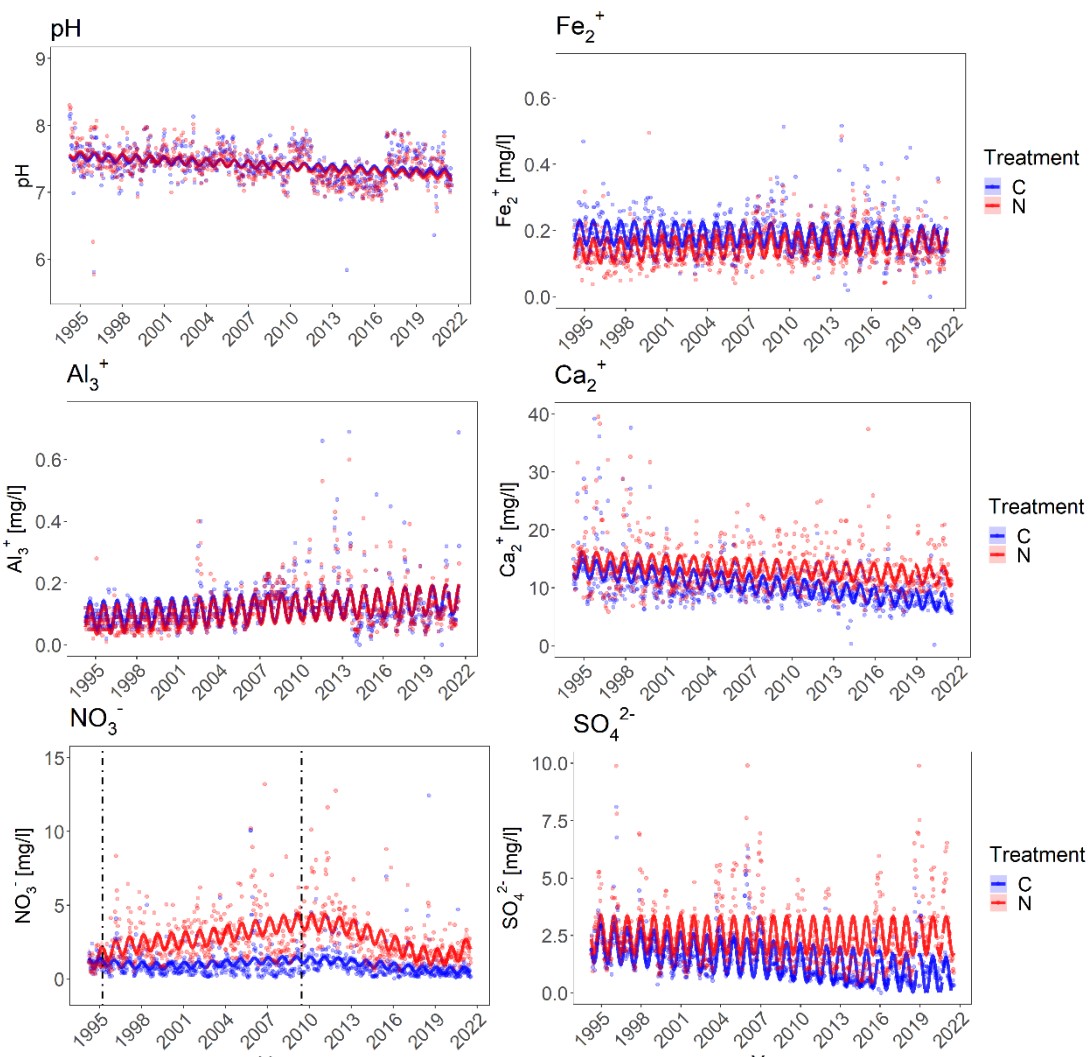

**Figure 6: Cation and anion concentrations in soil runoff water from the control (blue) and N-treated (red) catchment as a function of time. The points are measured values with a time step of 2 weeks. The continuous solid lines are the predicted values from the linear models with confidence intervals. 4 highest points were excluded from pH plot for O horizon. For the $NO_3^-$ a $5^{th}$ order polynomial was included in the lm model to capture the non-linearity of the data. The first dashed line indicates the beginning of the addition of $NH_4NO_3$ to the N-treated catchment. The second dashed line indicates the date of ringing of trees prior to felling. For $Fe_2^+$, $Ca_2^+$, $NO_3^-$ and pH, 1 outlier was excluded each from the plot and for $SO_4^{2-}$ 4 outliers were excluded.**