# Peer review of "Long-term additions of ammonium nitrate to montane forest ecosystems may cause limited soil acidification, even in presence of soil carbonate"

_Biogeosciences, 2023_

## Author Response (AR1)

**bg-2023-38**

We thank the associate editor for considering our manuscript, the comments of the reviewers and our answers to their comments. Below are our responses in blue, with the original editor's comments in black. We recall our answers to the reviewers (AC-1, AC-2) further down.

*Thank you very much for uploading your detailed responses to the comments by the two anonymous reviewers.*

*The evaluation and comments by the reviewers were positive and it seems like you are willing to address these comments on a revised version of your manuscript.*

*I am particularly interested in seeing how you addressed the following important issues:*

*Reviewer #2 made a very pertinent comment on what would be the impact of temporal autocorrelations among trends of pH over time in different compartments of the hydrological system. This analysis is particularly relevant to your study given its long-term and large-spatial scale.*

Beside the cross-correlations between precipitation pH and runoff water pH mentioned in AC-2, we now performed also auto-correlations of these time series. Without real surprise, they mainly showed an annual cycle, which justifies including the seasonality in our trend analyses by regressions.

*In addition, reviewer #1 raises important questions regarding the impact of N deposition on the ecosystem. Again, given the long-term nature of this study I believe that it is important to address how this process could have influenced your results.*

Indeed, the question of the impact of N deposition is addressed throughout the manuscript. In the revised version, we no longer only concentrate on the acidifying effect. We recall shortly previous findings about effects of N as a nutrient as seen in this experiment. This is important for the readers in order to better understand the context of the original results presented here.

**AC-1**

We thank Reviewer 1 for carefully reading and commenting our manuscript. Below are our responses in blue, with the original reviewers' comments in black.

*The current manuscript addresses relevant scientific questions within the scope of the journal: testing the effects of increased N deposition on acidification processes in a well-buffered ecosystem. This is a concept not enough explore so far, which gives high scientific relevance to the paper. The hypothesis are clear, well described, and fully addressed through methods and assumptions, that are clearly explained.*

We appreciate the reviewer's recognition of our manuscript's contribution to ecological research.

*- line 14-15, Although it is known from previous studies, the text here it is not clear enough about the N addition. Please clarify if N treatment is 22 kgN greater than the ambient N deposition (C treatment), that was 12 kgN at the beginning of the study, or if it represented 22 kgN compared to 12 kgN of ambient deposition.*

Indeed, these are 22 kg as treatment on top of 12 kg of ambient deposition. The abstract will be improved to make it clear that 22 kg is the additional input.

*- line 139-145: It could be also clarified a little.*

Especially the time course was not clear, with one-year of measurements preceding the start of the N treatment. The text will be amended to explain what we mean with this calibration period.

***Deposition.*** *Although N deposition and cycling is not key for the study, some further data a discussion on this issue could improve the understanding or, at least, allow further interpretation of the results. Particularly, the nitrate concentration response in runoff could be related to N cycling dynamics that can be connected to saturation processes. Knowing that N leaching is addressed in previous studies, not much information would needed here, but some of it would be appreciated. Particularly, a historic evolution of N deposition comparing both treatments would be appreciated. Moreover, in this study, throughfall deposition (if available) would be of greater relevance than bulk deposition since it represents a direct input into the soil.*

It is certainly a good idea to recall here the main results obtained about nitrate leaching. In this regard, we can distinguish three main periods over the course of our experiment: (1) immediate response due to preferential water flow through the soil, (2) progressive N saturation and (3) several years of increased leaching after girdling, then felling part of the trees. These results are already published in more detail but will shortly be recalled when we discuss nitrate leaching in the present contribution.

As it also partly represents dry deposition, throughfall is indeed a valuable information. It is already given in the material and methods and will be added to the introduction and also to the abstract.

*- line 15, line 92, line 130: the deposition value referred here is from the beginning of the study*

N deposition did not change much over the course of the experiment, but it is indeed better to write explicitly that the given numbers are from the beginning of the experiment.

*- line 130: total deposition and N retention in the catchments are also available from previous studies. The authors might consider if these data could be relevant here.*

This point is also justified and can be answered in the same context as the 3 previous ones.

*- lines 273-280: deposition data from other studies are referred here. Since it is stated that precipitation has not significantly changed in the valley, the changes in N concentration in rain are correctly assumed to be reflected in deposition values. However, some data on measured deposition trends would be appreciate here.*

Indeed, even if the trend in precipitation in not statistically significant, it still slightly modifies the trend of the fluxes compared to the concentrations. We will include these trends in Tab. S1.

***Nitrate leaching.*** *lines 297-299: This is just a question. It is stated that "$NO_3^-$ can have an acidifying effect if leached together with base cations, leading to their replacement by $H^+$ or $Al^{3+}$ on the exchange complexes". This effect is observed in N treatment in the increase of $NO_3$ leaching, that is related to the strong increase in soil acidity on the mounds. Should it be also related to an increased leaching of base cations that matches the $NO_3$ leaching but that cannot be observed in the current graphics?*

$Ca^{2+}$ is the dominant cation in runoff water. It can be seen (Fig. 6) that $Ca^{2+}$ concentrations decrease more in the control catchment than in the N-treated catchment. Additional $NO_3^-$ leaching from the treated catchment can partly (only partly) explain this difference in $Ca^{2+}$ trends. A short indication about this will be added here in the discussion.

***Figure 1.*** *I understand that trenches in catchment 1 are present in the same way as in catchment 2 (around the entire perimeter) but it is not drawn in the picture.*

On its upper part, catchment 1 (control) is delimited by a natural water divide. As the contour lines drawn on the graph are barely visible, a short explanation will be added to the legend of the figure.

***Please consider the following suggestions.*** *(Detailed comments not repeated here)*

All these comments are useful and will be taken into account. Only the 4[th] one (about decomposition) does not contain any particular action to be taken. The sentence was thus just considered with a particular attention within the overall language editing of the whole manuscript.

**bg-2023-38-AC2**

We thank Reviewer 2 for carefully reading and commenting our manuscript. Below are our responses in blue, with the original reviewers' comments in black.

*The paper by Baer et al. monitored changes in rainfall, soil, and runoff chemistry (pH, acidity and ions) in the preAlps during more than 20 years during which actions were taken to abate atmospheric pollution in Europe. They also added N to plots to simulate an increase in N deposition. This is an important contribution to understanding the long-term effects of increased N deposition in forest catchments, as well as of the importance of abatements measurements. As such, the current study is a highly valuable contribution to understanding patterns of N deposition and its consequences for the biogeochemistry of European forests.*

We much appreciate the overall positive assessment of the reviewer.

*On the negative aspect, more care should have been taken to avoid incomplete sentences, colloquial expressions, and grammatical errors and typos. Thus, the paper should be carefully revised in this aspect.*

The manuscript will be given a thorough revision to correct for grammatical errors, typos, redundancies and incomplete sentences while avoiding and replacing colloquial expressions. The text will also be revised by a third party (non author) competent in both English language and scientific writing.

*Moreover, some questions remain untested. For example, one of the most intriguing results is the increase in the pH of rainfall across time, indicative of reduced acid rain due to political action, but the continuous decrease of soil pH and pH in runoff water across the same time period.*

It is indeed not an obvious but an important result. In the discussion we consider the cause-to-effect relationships and their timing. It is especially important to distinguish between fluxes and buffering mechanisms. We could add here in a very short manner: it is not because there are less acidifying inputs that there is no longer any acidification. The discussion will be made more explicit in this regard.

*I wonder whether there is any temporal correlation among these variables. Or in other words, are shifting pH values across time in rainfall, soil and runoff coupled or decoupled? Are these couplings/decouplings maintained across soil horizons and experimental treatments? The same can be asked for other variables like total acidity, etc. Analyzing these types of questions would, in my opinion, increase the novelty of the paper beyond what could be considered as a very informative and highly valuable report by providing more novel insights.*

As shown by our results, precipitation pH, soil pH, soil exchangeable acidity and runoff pH have each a different time-course over the duration of the experiment. At a multi-year time-scale, they are clearly decoupled. Nevertheless, we gladly retain the reviewer's question to examine whether pH measurements in precipitation and in runoff water are correlated in the shorter term. After excluding the cold-season data that may be affected by a snowpack and after removing the long-term trends, we did not find any significant short-term correlation. This is not really surprising if we consider the large buffering capacity of the soil. We will add this new aspect to the revised manuscript. In terms of soil pH and exchangeable acidity, we have too few time points to try any such statistical approach and have to stay with the contrasting long-term trends of these time-series.

*Other than that, I consider this paper as a very valuable contribution to the field. (Detailed comments not repeated here)*

Most of these suggestions will be directly implemented. It is true that chemical equations are not commonly shown in ecological research articles. Nevertheless, we consider them useful because readers in this field often don't remember exactly what these reactions are and which (quantitative) consequences they have.
The sentence on lines 168-171 is about the replicated plot design. It is always a bit difficult to explain that we used two statistical designs for a single experiment: a paired-catchment design (with the advantage of being able to measure export fluxes in runoff water) and a replicated block design (with the advantage of the replications). We believe that everything will be more understandable by adding here a reference to Fig. 1, on which both designs are visible.

---

## Author Response (AR2)

**bg-2023-38**

We thank the associate editor for checking again our manuscript and for her new constructive comments. Below are our responses in blue, with the original editor's comments in black. The new line numbers refer to the new version with modifications tracked.

*After reading and evaluating the manuscript for the second time, there are two things that are not clear to me yet. First, I am not sure whether the griding and felling treatment was an explicit part of the experimental design. It appears that this was an intentional treatment that was replicated in the control and in the experimental plots. Also, it seems like the results of this treatment were presented in detail in another study, but here it appears that this experimental treatment still had a very significant effect on the acidity of not only soil, but also of runoff water. However, this does not become clear until the reader finishes the discussion. Therefore, it might be necessary to add a section in the introduction explaining how reducing basal area would have a feedback mechanism on soil and runoff properties derived from decreased N demand from plant uptake and decreased litter input. These mechanisms are invoked in the discussion without having been introduced properly. In addition, the authors would need to consider incorporating this treatment explicitly in their experimental design and statistical analyses.*

Girdling and felling were a supplementary treatment done on both the N-addition catchment and on the control (rainwater only) catchment. This supplementary treatment had a strong effect on nitrate leaching and this was the main result in Schleppi et al. (2017, cited in the manuscript). Girdling and felling may have had a transient effect on the pH of discharge water by reducing the acidification due the exchange with protons as roots take up base cations (L. 405). $Ca^{2+}$ in discharge water is, however, much more concentrated than $H^+$, which means that changes due to reduced root uptake may affect $H^+$ in runoff water without being visible for $Ca^{2+}$. It seems that there was a misunderstanding between this aspect and the difference in $Ca^{2+}$ trends between the N-treated catchment and the control. The duration and the intensity of the N addition are well sufficient to induce such an effect. We thus write that the significantly different trend observed in the N-treated catchment compared to the control may be related to the acidifying effect of the N addition. Indeed, more acidity means more dissolution of limestone and more release of $Ca^{2+}$. As a buffering process it affects more $Ca^{2+}$ fluxes than the resulting $H^+$ concentrations. Buffering is described in the introduction and barely requires more explanations here in the discussion. Still, we need to make it clear that the discussion about $Ca^{2+}$ in discharge water relates only to the N treatment and not to girdling and felling (in contrast with for nitrate leaching). This was done by recalling which treatment and which process is meant in each case (L. 357 and 405).

*Second, the experimental design and measurement methodologies are not explained clearly. It is not clear why while there were two catchments, the experimental plots where rainwater (with or without extra N) were outside these catchments. The spatial details and chronology of the study are not explained clearly. (…) Also, why are there experimental pairs of plots outside the monitored catchment? And why are there no experimental pairs of plots inside catchment two?*

It is true that we introduced the replicated plot design too late in the text. As suggested, we write now that we used a dual experimental design right at the beginning of the description of the N-addition experiment (section 2.2, L. 137-147). We now refer already here in the text to Fig. 1, which clearly shows how both designs lie together in the field . We clarified also the meaning of the symbols for the replicated plots in the legend of that Fig. 1. Only after this basic information about the dual design do we bring details about the treatment itself. We believe that this description from more general towards more detailed will indeed be preferable for the readers.

*Furthermore, there seem to be some inconsistencies in how water and soil samples were measured. The information regarding the analytical methodologies is hidden in different sections of the methods, but I think that in this study this is critical information and deserves a sub-section of its own. Also, in this sub-section, the authors should clarify potential changes in instruments or methodologies that were (understandably) implemented over the course of such a long-term experiment.*

There were indeed some inconsistencies in the soil analyses as we relied mostly on samples dried soon after collection, but in one case we used samples that had first been frozen. This is acknowledged in the text (L. 197). There was, however, no specific result that could be linked to this fact and would have needed discussion. Laboratory analyses were done with standardized methods regularly checked with international standards. Otherwise, there were only potential inconsistencies due to aging of the pH-meters and renewing of electrodes, in spite of calibrations before each measurement series. As explained in the text (L. 180 ff.), a long-term trend due to the instruments was checked by remeasuring together samples that had been frozen after collection along 21 years. The trends observed on fresh samples or on samples frozen and later measured together were not equal but not really significantly different either (L. 328-330). We show thus both regression slopes, and now also their standard errors (L. 256 and 260). We discuss also possible errors that may arise from the conservation of frozen samples (L. 332-336). It is true that uncertainties remain about the exact rate of change and we do not hide this at all. Still, the main fact is clear: a strong increase in pH over time.

Because (potential) inconsistencies are so different from each other, we do not consider it useful to gather them into a specific section but prefer to discuss them directly along with the corresponding methods and results.

*In line with these comments, one wonders whether the observed decrease in pH of runoff water in control plots would be simply due to the addition of extra rainwater, which already has a baseline concentration of N forms. How do these trends compare to similar measurements of pH in runoff water from similar catchments?*

Compared to the N addition of 22 kg ha$^{-1}$ y$^{-1}$ in the treated catchment, adding ambient rain water to the control catchment corresponds to 0.8 kg ha$^{-1}$ y$^{-1}$. It is very unlikely that this small amount cannot induce almost as much pH decrease that the amount sprayed to the treated catchment. The difference in slope in Fig. 6 is indeed very small. However, we take the chance of this question to add in the text that the amount of precipitation water added represents 7% of the ambient precipitation, allowing the readers to immediately see the order of magnitude that it represents, also in term of acidifying inputs (L. 139).

In the discussion, we now compare our trend for pH in the forested catchments with a third nearby catchment (L. 414-416). This third catchment is fallow land, originally mostly abandoned grassland, now with more and more trees. Due to this slow change in vegetation, it is not really a good comparison for pH trends, but it is the only comparison that we have to answer this specific question. Therefore, we indicate the trend observed there without going into more details about the own dynamics of a fallow land, which would be a separate topic.

*In addition, I provide some more comments that I hope they will improve the clarity of the language, which can still be polished. Comments that I consider are more relevant are separated by an extra line space. Line numbers (L) refer to the version with track changes.*

*L11 N deposition is not exclusive to forests. Also "forest ecosystems" can be shortened to "forests".*

Done (L. 11).

*L12-13 what do you mean by "sensitive receptors of the environment".*

Critical loads (CL) are generally defined in two ways: empirical CL that are based on vegetation changes and mass balance CL that consider the budgets of chemical elements. We indicate now these two aspects in a broad sense: "biodiversity and ecosystem functioning" (L. 12).

*L20 name these "soil acidification processes".*

We removed the word "processes", as we examine the soil acidification itself and not the processes (L. 19).

*L20-21 I do not understand the addition of the clause "while previous reports…" I think you could remove this from the abstract*

Done (L. 18).

*L23 the sentence "In spite of…" is repetitive, this same idea is given in the previous sentence.*

Done (L. 21).

*L23 Remove "First".*

Done (L. 22).

*L25 what does "they" refer to here?*

We mean the mounds, not explicitly (L. 23).

*L31 Close your abstract with a brief conclusion or future perspective derived from your study.*

We included the main outlook from the conclusion into the abstract (L. 29-31).

*L33 remove "more than".*

Done (L. 34).

*L36 more concise language: remove "are a" and change "of" by "from"*

Done (L. 37).

*L38 more concise language: "and deposited in dry or wet form".*

Done (L. 39).

*L40 New paragraph starting with "As large regions…"*

Done (L. 42).

*L54 provide a citation for this.*

This sentence is probably correct considering all the decomposition processes, but it is a sum of processes consuming protons and other producing protons. We did not find any study that would be broad enough to really conclude that consumption generally dominates. Therefore, we decided to remove the sentence.

*L62-64 I am not sure this is very relevant for this study and you could consider removing it.*

Done.

*L116 "at a site"*

Done (L. 110).

*L125 "a deeper water table" instead of a "lower laying water table" (unless you meant a "more shallow water table", which is not entirely clear here).*

Done (L. 119).

*L130 specify the net rate of which process.*

It's about nitrification, now explicitly (L. 124).

*L136 remove "their".*

Done.

*L146 & 179 provide manufacturer details.*

The weirs are a self-construction and this is now written so (L. 154). The manufacturer of the pH-meter and its location are now both given (L. 177).

L141 Here is where you should present the experimental design in detail and how the treatments were applied (currently L180-191)

See the answer to the second general comment above.

L140 I assume these values of N deposition refer to the levels at the beginning of the experimental treatment, please clarify this.

Yes, now clarified (L. 133).

L142 According to the responses to the reviewers' comments, one of the catchments was not delimited by a trench, could you clarify this?

This was only clear in the legend of Fig. 1, now it is also written in the text (L. 151).

L166-167 Can you provide the maker, model and manufacturer details of the rain gauges. Also, how did these gauges collected and measured precipitation? Were the two rain gauges located in the same open area? Where was this area with respect to the experimental and control plot?

The rain gauges were self-made from three parts: a tube, a funnel and a collection bottle. These details are now indicated (L. 172 ff.).

L174-177 How consistent were measurements taken more than two decades apart? How can you relate potential drifts with changes in measurement instruments, technology and their precision?

See the answer to the second general comment above.

L190 what does "it" refer to here?

Clarified (L. 200).

L192 what do you mean here by "in general"?

Deleted (L. 202).

L214 provide the corresponding package citations for this function.

Function acf in the core of R and there is no need for more indications.

L215-216 This belongs to the results section, also, what are "periods" here?

The periods are always of two weeks. This is now explicit in the text (L. 304-305).

L232 I do not understand this, I thought the variable "treatment" was included as a fixed factor on the models testing for the trends of soil acidity.

There was a problem with the order of the explanations that indeed made this confusing. We reordered the text to make it clear (L. 238 ff.).

L245 remove "highly" and "atmospheric" (precipitation is by definition atmospheric).

Done (L. 254).

L246 what do you mean here by "at the beginning of the project"? Maybe use instead "at the start of the experimental treatment" or "pretreatment measurements"?

Done (L. 254).

L247 remove "almost".

Done (L. 255).

L247 & 250 These rates of change should be accompanied by an error estimate.

Done (L. 256 and 260).

L248 what do you mean here by "weather events"?

This is now clarified (L. 258).

L251 where do you show this result? Please refer to the corresponding table or figure.

Reference to Tab. S1 and Fig. 2 added here (L. 261).

L263-269 replicates instead of replications. Also, consider writing your results in past tense.

Done (L. 272 ff.).

L269 "were the most acidic" instead of "have the highest acidity".

Done (L. 277).

L307 add "of precipitation water" after $NO_3^-$.

Done (L. 317).

L319 pH instead of "it".

Done (L. 327).

L325 remove "likely"

Done (L. 334).

L333-334 I am not sure I understand this, soil acidification has to be intrinsically linked to the concentration of $H^+$.

It is about the effect of the acidity in precipitation on soil acidification. This is now clarified (L. 341).

L339 remove "in greater detail".

Sentence rewritten without these words (L. 348).

348-349 I am not sure I follow how decreased N demand by the trees has a feedback effect on the concentration of Ca+, could you explain this?
L349 But I thought the felling resulted in a decrease in N demand, this is contradictory.

See the answer to the second general comment above.

L350-351 Please provide a citation to support this claim of low nitrification rates.

A reference is now given (L. 360).

L364 what soil property is buffered?

Done (it's about cation exchange, L. 372).

L371 provide citations and examples. Also consider "lack" instead of "don't contain".

We give now a reference about the history of research on this topic as well as a study centered on Switzerland, the country of our own experiment (L. 379).

L376 how would deeper rooting help buffer acidification due to increased N deposition?

Deeper rooting would facilitate the uptake of base cations from soil horizons, which sustains the buffering by cation exchange. This is now clarified in the text (L. 384 ff.).

L376-379 This line of argumentation linking plant litter, denitrification and cation transport between moulds and depressions is not clear.

Trees (both root systems and crowns) are larger than single mounds or depressions. Uptake of nutrients and their restitution to the soil as litter act thus as a redistribution of elements between the small topographical features. This is now clarified in the text (L. 384 ff.).

L383 "peaked"

Done (L. 396-397).

L416 "emissions of N and S oxides"

Done (L. 433).

Figure 1. Add a legend indicated what are the blue and red circles.

We added this in the legend of the figure (rather than a separate legend within the map because it would in our opinion add complexity).

Figure 2-6. Please, increase font size of all labels and axes.

Done.

---

## Author Response (AR3)

**bg-2023-38**

We thank the associate editor and the editor for the few more comments on our manuscript.

Concerning the first one:

*L115-136 Consider mentioning here the felling experiment, when it took place and how this treatment did not have any significant impacts on the current study.*

We moved the corresponding paragraph (now L. 131-136 in the final version). Only the fact that girdling and felling were quantitatively similar on both catchments had to remain after the text introducing the catchments themselves (L. 154-156). And the sentence about ambient deposition that was at the end of section 2.1 now fits better further up (L. 123-125), along with the climate and not after the sentences about vegetation.

The other two small corrections wished by the associate editor were done as wished.

Finally, as wished by the editor, we improved Fig. 1 for people with color blindness and also for the case that it is printed in a gray tones. (The contrast between the circles representing the circular plot was indeed not optimal.)